# *Anaplasma* Species in Ticks Infesting Mammals of Sardinia, Italy

**DOI:** 10.3390/ani13081332

**Published:** 2023-04-13

**Authors:** Valentina Chisu, Silvia Dei Giudici, Cipriano Foxi, Giovanna Chessa, Francesca Peralta, Valentina Sini, Giovanna Masala

**Affiliations:** Dipartimento di Sanità Animale, Istituto Zooprofilattico Sperimentale della Sardegna, Via Vienna 2, 07100 Sassari, Italy

**Keywords:** *Anaplasma*, ticks, vector, tick-borne disease

## Abstract

**Simple Summary:**

Bacteria in the *Anaplasma* genus are intracellular parasites of mammal blood cells transmitted by ticks of genera *Amblyomma*, *Dermacentor*, *Hyalomma*, *Ixodes*, and *Rhipicephalus*. In this study, the presence of *Anaplasma marginale* and *A.*
*phagocytophilum* in ticks was molecularly confirmed in *Rhipicephalus sanguineus* s.l. and *Rhipicephalus bursa* ticks, suggesting that these tick species are of importance in the transmission of potential zoonotic infections. Due to the nature of the *Anaplasma* species detected here, our results, together with data obtained to date in Sardinia, suggest that from a public health point of view, the potential zoonotic *Anaplasma* species should be further investigated in the island. Additional studies are needed to clarify whether these tick species can transmit these zoonotic bacteria both to human and animal hosts.

**Abstract:**

Ticks are hematophagous ectoparasites that are recognized for their ability to vector a wide variety of pathogens of viral, bacterial, protozoal, and helminthic nature to vertebrate hosts. Among the different diseases transmitted by ticks, also called “Tick-Borne Diseases” (TBD), many are zoonotic. Pathogens of the genus *Anaplasma* refer to obligate intracellular bacteria within the Rickettsiales order transmitted mainly through tick bites and considered as well-established threats to domestic animals, livestock, and humans, worldwide. In this retrospective study, 156 ticks collected from twenty goats, one marten, and one cattle from several Sardinian sites, were examined by molecular analyses to detect the presence of *Anaplasma* species. A total of 10 (10/156; 6.4%) ticks were shown to be *Anaplasma*-positive by PCR screening. After sequence analyses, *A. phagocytophilum* was detected in four *Rhipicephalus sanguineus* s.l. (3.3%) and four *Rh. bursa* (11%) ticks from goats, while one *Rh. sanguineus* s.l. (0.8%) and one *Rh. bursa* (2.8%) collected from the marten and cattle, respectively, exhibited 100% of identity with *A. marginale* strains. In this study, we provide the first description and molecular detection of *A. marginale* and *A. phagocytophilum* in ticks of the *Rhiphicephalus* genus in Sardinia. Considering the growing impact of tick-borne *Anaplasma* pathogens on human health, further studies are necessary to monitor the prevalence of these pathogens in Sardinia.

## 1. Introduction

Ticks are hematophagous ectoparasites that transmit protozoan, bacterial, and viral pathogens of medical and veterinary importance [1]. Among bacterial pathogens, members belonging to the genus *Anaplasma* (Rickettsiales: Anaplasmataceae) are obligate intracellular organisms that replicate within parasitophorous vacuoles in the cytoplasm of both vertebrate and invertebrate host cells [2].

In the vertebrate hosts, these organisms infect blood cells including erythrocytes (RBCs), monocytes, platelets, and neutrophils and constitute a major public health threat in humans and animals [3]. The different *Anaplasma* spp. exist in a zoonotic cycle that involve both Ixodidae ticks and vertebrate hosts, which can be reservoirs of infection [4]. Trans-stadial transmission from nymph to adult exists in nature while transovarial infection has been suggested only for few Ixodidae species. Transmission of these pathogens occurs due to the action of ticks during their blood meal on infected animals and they can transmit the agent to other mammals at the next stage. Although *Anaplasma* spp. are mainly transmitted by tick bites, other modes of transmission such as hematophagous insect bites and exposure to blood-contaminated fomites have been proven [5].

The genus *Anaplasma* has been especially studied for its pathogenicity in farm animals since various species of *Anaplasma* such as *A. marginale*, *A. ovis*, and *A. bovis,* along with *A. phagocytophilum,* are regarded as one of the biggest threats to livestock [6]. In fact, these agents significantly affect animal productivity and cause considerable economic losses to farmers due to the reduction in reproductive performance, decreased milk and meat production, abortion, and death of the infected animal [7]. However, the clinical identification of infected animals is challenging because immunocompetent hosts do not exhibit symptoms while clinical signs ranging from subclinical infections with mild to high fever, anorexia, and respiratory symptoms have been described in immunocompromised hosts [8]. Although anaplasmosis is rarely fatal, leukopenia and impaired neutrophil and lymphocyte function of the bacteria can make animals more susceptible to life-threatening secondary infections [8].

Currently, there is a high diversity of *Anaplasma* organisms, which includes six validated species (*A. phagocytophilum, A. marginale*, *A. centrale*, *A. ovis*, *A. bovis*, and *A. platys)*.

Moreover, the genus contains two new species, namely *A. odocoilei* and *A. capra, *that have not been cultured yet**, as well as species of ‘Candidatus’ status and many other new unclassified *Anaplasma* genovariants that have been recently detected [9].

Among them, several species that were previously considered non-pathogenic were recently found to be zoonotic and associated with human diseases, suggesting that the number of *Anaplasma* species that are infecting humans is increasing. Specifically, *A. phagocytophilum*, the causative agent of tick-borne fever in sheep and granulocytic anaplasmosis in dogs (CGA) and horses (EGA), is also responsible for human infection (HGA) [9]. *A. bovis*, previously found to infect bovine monocytes, has been recently detected in humans in China [10]. *Anaplasma platys*, which infects platelets and is the etiological agent of infectious cyclic thrombocytopenia in dogs, has been documented in two women from Venezuela who were exposed to *Rhipicephalus sanguineus* [11]. A variant of the erythrocytic anaplasmal *A. ovis* was identified in a Cypriot patient with clinical signs including fever, hepatosplenomegaly, and lymphadenopathy [12]. Finally, *Anaplasma capra*, a novel, tick-borne pathogen which was detected in China in 2010–2012, causes zoonotic infections and infects many different animal species, including humans [13].

*Anaplasma* distribution is correlated with the presence of tick vectors, hosts, and competent reservoirs. Therefore, determining the density of ticks and the incidence of the infectious agents they transmit is important to prevent and avoid the transmission of possible diseases to animals and humans. Sardinia is the second biggest island in the Mediterranean Sea covering a surface of 24.090 km^2^ and with different habitat types. It has an annual mean temperature of 22 °C and a typical Mediterranean climate that allows the survival of ticks during the whole year. Furthermore, the island, which is located approximately halfway between Spain, Italy, and North Africa, is an important stopover area for migratory birds which pose a risk for the introduction and dispersal of ticks and TBD. Specific studies of the prevalence of *Anaplasma* spp. in Sardinian ticks are limited. It was previously observed that ticks belonging to the *Rhipicephalus *and* Hyalomma* genera are the most frequent hosts for *A. ovis, A. phagocytophilum*, *A. platys*, and *A. platys*-like, suggesting that these species could serve as potential vectors for these pathogens [14].

The aim of this study was to verify the distribution of *Anaplasma* species in ticks collected from mammals from Sardinia, Italy, and provide epidemiological data to develop strategies and control programs for anaplasmosis prevention and monitoring in the island.

## 2. Materials and Methods

### 2.1. Tick Collection

In this retrospective study, 156 tick specimens opportunistically removed between June 2011 and October 2013 from 20 goats, 1 cattle, and 1 marten in Sardinia (Italy) were analyzed for the detection of *Anaplasma* species. The sites belonging to Ogliastra and Sassari provinces were randomly chosen and ticks collected from this geographic area were included in this study. Ticks were removed from each host with tweezers and placed in vials containing 70% ethanol at room temperature. Ticks from one marten that was found dead were provided by hunters who removed specimens from the animal. Morphological identification of the ticks was conducted down to the species level using identification keys [15] with a binocular microscope at a magnification of 50×. Ticks were also sorted by stage and animal host and then stored at −80 °C until further analyses. Details of collection sites, species, and the sex of each tick were collected. The origins and hosts from which each tick was sampled are summarized in Table 1.

### 2.2. DNA Extraction and PCR

To remove environmental contaminants, the ticks were rinsed with 70% ethanol and then immersed in deionized water to remove the ethanol. The ticks were then longitudinally cut in two equal parts using sterile instruments for each individual dissection, and one half was used for DNA extraction. The half tick was homogenized with a Tissue Lyser (TissueLyser II) in 200 μL of PBS. One hundred microliters of genomic DNA was extracted using QIAgen columns (QIAamp tissue kit; Qiagen, Hilden, Germany, cod.69504), according to the manufacturer’s instructions. PCR amplification of the 16S ribosomal RNA gene was carried out on all genomic DNA samples by using oligonucleotide primer pairs (Eurogentec, Seraing, Belgium), which amplified a 293-bp fragment of *Anaplasma* species [16]. All reactions and amplification conditions used in this study were confirmed from studies previously published [14]. Negative and positive controls were included in each amplification assay. Eight microliters of each 293-bp PCR product was electrophoresed in 1.5% agarose gel with SYBR™ Safe DNA Gel Stain (Invitrogen, Carlsbad, CA, USA) in one × TAE buffer against a DNA ladder. The gel was then visualized and photographed using Alliance LD2 gel documentation system (UVITEC, Cambridge, UK).

An additional PCR was performed on positive DNA tick samples using 16SANA-F (5′-CAG AGTTTG ATC CTG GCT CAG AAC G-3′) and 16SANA-R (5′-GAGTTT GCC GGG ACT TCT TCT GTA-3′) primers that amplify 16S rRNA gene of *Anaplasma* spp., as reported in De la Fuente et al., 2005 [17]. The reaction was made up to 25 μL containing 12.5 μL of 2× PCR Master Mix (Quantitect Probe PCR Master Mix; Qiagen, Hilden, Germany), 1 μL of 25 μM of each primer, and 1 uL of template DNA. DNA extracted from uninfected ticks and DNA previously extracted from *A. phagocytophilum* IFA slides were included in each PCR test as negative and positive controls, respectively. Thermocycler conditions were performed in automated DNA thermal cyclers (GeneAmp PCR Systems 2400 and 9700; Applied Biosystems, Courtaboeuf, France) with the cycling conditions as follows: 95 °C for 15 min, 40 cycles of 94 °C for 30 s, 60 °C for 30 s, and 72 °C for 1 min, with the final elongation step at 72 °C for 5 min. The amplicons were then subjected to electrophoresis in a 1.5% of agarose gel at 110 V for 30 min and visualized using a Syber safe nucleic acid staining solution, under UV light.

### 2.3. Purification, Sequencing, and Phylogenetic Analyses

The *Anaplasma* positive samples were selected and purified using the QIAquick Spin PCR purification kit (Qiagen, Hilden, Germany) following the manufacturer’s instructions. The purified PCR products were then sequenced using the 16S primer pairs in both directions on an automated DNA sequencer (ABI-PRISM 3500 Genetic Analyzer; Applied Biosystems, Seevetal, Germany). The DNA sequencing kit (dRhodamine Terminator Cycle Sequencing Ready Reaction; Applied Biosystems) was used according to the manufacturer’s instructions. Chromatograms of nucleotide sequences generated in this study were assembled and edited with ChromasPro software (version 1.34; Technelysium Pty Ltd., Tewantin, Queensland, Australia). Sequences were then aligned with CLUSTALX [18] due to assign them to unique sequence types and checked against the GenBank database by using BLASTn analysis tool [19]. Pairwise/multiple sequence alignments and sequence similarities were calculated using the CLUSTALW [20] and the identity matrix options of Bioedit [21], respectively. The phylogenetic tree was constructed using the neighbor-joining method in MEGA software version 6.0. The distance matrix was calculated by use of Kimura-2 parameters. The statistical analysis was performed using the Bootstrap method with 1000 replicates. The method used to calculate a confidence interval for a proportion is the Wilson score method without continuity correction [22].

## 3. Results

### 3.1. Tick Collection

Ticks were morphologically identified at the species level as *Rh. sanguineus* s.l. (120 specimens) and *Rh. bursa* (36 specimens). Although *Rh. bursa* is not included in the *Rh. sanguineus* group, the species has been differentiated by the shapes of adanal plates for males and by the genital aperture, porose areas in the dorsal surface of basis capituli, spiracle plates, and the presence of dense setae in the spiracle areas for females. *Rhipicephalus bursa* shows adanal plates guttiform with maximum width at the posterior margin and with quite convex and divergent lateral margins and obtuse and broadly rounded posterior inner angles. *Rhipicephalus sanguineus* presents adanal plates with rectilinear or weakly inclined posterior margins and posterior inner angles almost right. The genital aperture posterior lips of *Rh. bursa* have a narrow “V” shape with divaricate and slightly rounded lateral margins, while *Rh. sanguineus* shows a genital aperture like a broad “U” with divergent lateral margins. Porose areas are nearly circular with a broad distance separating them in *Rh. sanguineus* and are oval with a narrow separation in *Rh. bursa*. *Rhipicephalus sanguineus* shows spiracles plates with narrow tails and the presence of sparse setae in spiracles areas while *Rh. bursa* have spiracles plates with broad tails and dense setae in spiracles areas. All ticks were adult specimens and no larvae and nymphs were removed from collected animals. Tick species and number, host source, collection sites, stage, and sex of ticks collected from this study are shown in Table 1.

### 3.2. Molecular Detection of Anaplasma spp.

A total of 10/156 tick samples (6.4%; 95% CI 3.5–11.4) tested positive for *Anaplasma* DNA using 16S rRNA PCR (Figure 1).

To molecularly determine the identity of *Anaplasma* spp. detected in ticks from this study, the PCR products of the 10 PCR-positive samples were directly sequenced.

A total of four sequences from *Rh. sanguineus* s.l. (3.3%; 95% CI 1.3–8.3) and four from *Rh. bursa* ticks (11.1%; 95% CI 4.4–25.3%), all collected from goats, were readable and chromatograms generated a clear sequencing signal containing an *Anaplasma* that showed 100% similarity with the 16S ribosomal RNA fragment of *A. phagocytophilum* strains after BLAST search analyses.

One *Rh. sanguineus* s.l. from marten (0.83%; 95% CI 0.15–4.57) and one *Rh. bursa* tick from cattle (2.78%; 95% CI 0.49–14.17) contained *Anaplasma* DNA that exhibited 100% similarity with *A. marginale* strains deposited in GenBank. All 16S rRNA PCR product sequences resulted in the establishment of two 16S different genotypes, as shown in Table 2.

Specifically, eight sequences from ticks sampled from goats were identical to each other and to *A. phagocitophilum* strains from GenBank, while two sequences from two ticks collected from one marten and one cattle were identical to *A. marginale* strains isolated worldwide (Table 2). The two different sequence types generated in this study named AP-SAR2011 (*Anaplasma phagocitophilum* sequence type) and AM-SAR2011 (*Anaplasma marginale* sequence type) were deposited into GenBank under the accession numbers KP877313 and KP877314, respectively (Table 2). The phylogenetic analysis based on the partial 16S rRNA (Figure 2) showed that the *A. phagocitophilum* strain AP-SAR2011 found in this study was in the same clade as strains isolated in South Korea, Russia, and China from humans, ticks, rats, and goats. The *A. marginale* strain AM-SAR2011 was in the same clade as the Italian strain BS16, isolated from bovines, and close to the strains isolated in Philippines and South Africa.

## 4. Discussion

The rapid identification of different tick species and the bacteria they carry contributes substantially to the clinical diagnosis, treatment, and surveillance of tick-borne diseases. Current knowledge of the incidence of *Anaplasma* species in Sardinian ticks is limited, and obtained results, based on specific molecular typing, highlighted that *Rhipicephalus* and *Hyalomma* ticks harbored several species of *Anaplasma* (*A. ovis*, *A. platys-like*, *A. platys* and *A. phagocytophilum*), raising concerns regarding their potential to transmit these pathogens to humans, domestic hosts, and wildlife. The presented data show that within the five monitored sites, two different tick species were identified. Specifically, *Rhipicephalus sanguineus* s.l. ticks were the predominant identified species documented here, and this result was consistent with data from the previous literature [23]. *Rhipicephalus* species are widely distributed across the Mediterranean region, as well as in Sardinia where the *Dermancentor*, *Haemaphysalis*, and *Hyalomma* genera are also well represented and adapted to the ecosystem [24]. In this study, the presence of *Anaplasma* species was recorded in 6.4% of tested ticks. It does not mean that these ticks are competent vectors for the bacteria, as the ticks may have been infected by feeding on bacteremic animals or by cofeeding with tick vectors.

In particular, the obtained results indicated that, like previous studies in which *A. marginale* was detected in *Rh. bursa* and *Rh. sanguineus* s.l. ticks from Italy, Portugal, and Spain [25,26,27], Sardinian tick species are hosts for *A. marginale*, suggesting that both tick species could serve as potential biological vectors for *A. marginale* infection. To the best of our knowledge, this is the first molecular evidence of *A. marginale* in *Rh. bursa* and *Rh. sanguineus* s.l. ticks collected from goats and a marten in Sardinia, a region in which species of the *Rhipicephalus* genus are widely distributed.

*Anaplasma marginale*, the aetiologic agent of bovine anaplasmosis, represents one of the most important tick-borne diseases in ruminants worldwide, mainly in the tropical and subtropical regions. In the southern regions of Italy (Sicily, Puglia, Campania, and Basilicata), where bovine anaplasmosis is endemic, the presence of *A. marginale* in *Rh. turanicus* and *Haemaphysalis punctata* collected from cattle has been previously reported [27]. This agent infects circulating erythrocytes of domestic and wild ruminants [28]. Infected cattle serve as a reservoir of *A. marginale* providing a tick blood source for the efficient biological transmission of the pathogen [28]. Although approximately 20 tick species are reported as biological vectors of *A. marginale* worldwide [9], most of them are able to transmit *A. marginale* only under experimental conditions, which does not necessarily imply transmission in the field [29]. However, it has been demonstrated that the pathogen can be mechanically transmitted by blood-contaminated mouthparts of biting diptera of the *Tabanus* and *Stomoxys* genera, or via fomites [30]. In this study, ticks that tested positive for *A. marginale* were removed from asymptomatic goats. However, we cannot know the health status of the marten whose ticks tested positive for *A. marginale* since it was found dead for unknown causes. Results from a previous study conducted in northeastern Hungary highlighted the presence of *Anaplasma* sp. in spleen and liver samples of the European pine marten in which the zoonotic ecotype I of *A. phagocytophilum* has been identified [31]. More investigation on *A. marginale* in *Rhipicephalus* ticks and in domestic and wild vertebrate hosts could help to highlight the possible role of these ticks as vectors of *Anaplasma* species. Although phylogenetic analysis based on *Anaplasma* sequences obtained with 16S rRNA gene amplification revealed that this target gene can be widely used for the identification of *Anaplasma* species and can be considered a valuable phylogenetic tool, more discriminative genes will be used for the confirmation of these results.

This study also reports the first molecular detection of *A. phagocytophilum* in *Rh. sanguineus* and *Rh. bursa* ticks in Sardinia, indicating a potential role for these tick species in the epidemiology of the disease. *Anaplasma phagocytophilum* is the cause of granulocytic anaplasmosis in humans (HGA) [32], which severity ranges from asymptomatic infection to mild or severe febrile illness and involvement of multiple organ failure or even death [33]. Therefore, several genetic variants of this pathogen have been determined and all of these differ from each other for the different host specificity, vectors, pathogenicity, and geographical distribution [34]. In fact, all variants can infect different species as demonstrated by experimental studies in which it has been proven that *A. phagocytophilum* strains isolated from different matrices are not capable of infecting different hosts [4]. The analysis of this genetic variability has been made through molecular methods by using different loci such as 16s rRNA, groESL msp2, msp4, and ankA genes. Thanks to the use of one or more of these molecular markers, *A. phagocytphilum* has been divided into different genetic variants that can be involved in different epidemiological cycles, distribution, and host spectra [4]. Although 16s gene rRNA is the most used target gene, the phylogenetic study based only on the use of 16S rRNA could be deficient. It is related to the genetic recombination of this gene that undergoes several variations [4]. Moreover, if used alone, it may not be discriminative enough to correctly differentiate the different types of *Anaplasma* species, although different variants have been identified by using it as a reference point. The analysis of nucleotide sequences using the groESL gene has allowed for the identification of four different ecotypes of *A. phagocytophilum* in Europe, and different genetic variants adapted to the different hosts and vectors present in a specific geographical area [35]. All these ecotypes can infect both vertebrate and invertebrate hosts. In particular, the ecotype I has a wide host spectrum, being associated to multiple animal species including humans. This ecotype with zoonotic potential has the largest range in wildlife reservoirs but can also infect domestic animals. Specifically, hosts of this pathogen include cattle, sheep, goats, horses, dogs, hares, yaks, and rodents [36] and evidence of the pathogen in several mammalian and invertebrate hosts have been reported in Italy as well [37,38,39]. However, one limitation of this study was the lack of identification of *A. phagocytophilum* ecotypes, which is essential information for defining the zoonotic relevance. Further studies are needed to better characterize strains by analyzing more discriminative genes and to identify the main vectors implicated in the transmission of *Anaplasma* species in Sardinia.

Moreover, the *A. phagocytophilum* strain detected here was close to *A. phagocytophilum* strains isolated in China, Korea, and Russia. It could be related to anthropogenic activities that contribute directly or indirectly to the emergence and re-emergence of tick-borne pathogens (e.g., animal production, animal–human interfacing, and globalization). Moreover, the role of migratory birds in the spread of ticks and their role in the circulation and dissemination of pathogens in Europe cannot be ruled out. During seasonal migrations, birds that cover short, medium, or long distances within one or more distant geographical regions can carry ticks and related pathogens, introducing ticks and pathogenic species to new areas [40].

Although *I. ricinus* is the main vector of *A. phagocytophylum* in Europe [41], the zoonotic pathogen has been also associated with *Rhipicephalus* and *Dermacentor* spp. ticks from other parts of the world [42]. However, since the Mediterranean climate could be a limiting factor for *Ixodes* distribution and it would explain the low population of *I. ricinus* in Sardinia, the abundance of *Rhipicephalus* ticks in the Mediterranean basin prompted us to suggest that the *Rhipicephalus* species can serve as vectors of *A. phagocytophilum* and may transmit the pathogen to animal hosts. Therefore, the number of ticks was very low and they were collected from the hosts. Therefore, no conclusion can be drawn whatsoever about the circulation of the pathogens within the tick population, as every detection could be the result of infected ingested blood. These results confirmed the presence of *A.*
*phagocytophylum* in *Rhipicephalus* ticks feeding in goats. In Sardinia, where ruminant breeding represents a zootechnical reality of primary importance, the increase in the incidence of anaplasmosis transmitted by vectors could represent a serious threat to company profitability. According to the National Italian Database 2020 (BDN) (established by the Ministry of Health at the National Surveillance Centre of the IZS in the Abruzzo and Molise Region), Sardinia has an estimated population of more than 3 million sheep and 0.3 million goats, and losses due to abortion of ruminants are estimated to be around EUR 10 million per year. Veterinarians should not overlook the presence of *A. phogocytophilum* in Sardinian goats and evaluation of the potential role of *Anaplasma* species as abortifacient agents should be also taken into account. Improving the entomological surveillance program is necessary to establish and maintain a dialogue with farmers, including listening to and addressing their concerns and sharing an adequate diagnostic and therapeutic path for the good health of farm management through innovative solutions that will reduce the economic losses in this area and ensure the efficiency of vector control interventions. Furthermore, since the risk of transmission of vector-related diseases is extended to the entire population, particular attention must be paid to professional categories who carry out their activities outdoors.

## 5. Conclusions

The knowledge and characterization of the diversity of *Anaplasma* strains circulating in the island are fundamental to design epidemiological studies and control strategies for both HGA and bovine anaplasmosis. The results of this study showed that *Rh. sanguineus* s.l. and *Rh. bursa* harbor two *Anaplasma* spp., of which *A. marginale* was not yet reported in the territory, and support the hypothesis that these tick species could act as vectors for *A. marginale* and *A. phagocytophilum* in Sardinia. Further investigation to fully understand a possible role of *Rhipicephalus* ticks in the *A. phagocytophilum* and *A. marginale* strains circulation are warranted. Furthermore, future studies may yield more insight into the seroprevalence of *Anaplasma* species in ruminants and in the dog population of Sardinia considering their potential role in the transmission of the disease to humans.

## Figures and Tables

**Figure 1 animals-13-01332-f001:**
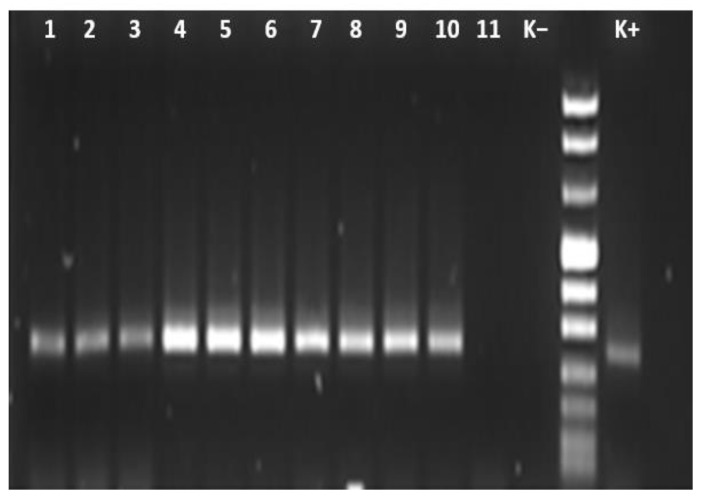
Agarose gel electrophoresis (1.5%) of PCR amplification products of *Anaplasma* spp. with 16S rRNA primer. Lanes 1–11: numbers of the strains; Lane 11: K(−)-negative control; Lane 12: DNA Size Marker (Marker VIII); Lane 13: K(+)-positive control.

**Figure 2 animals-13-01332-f002:**
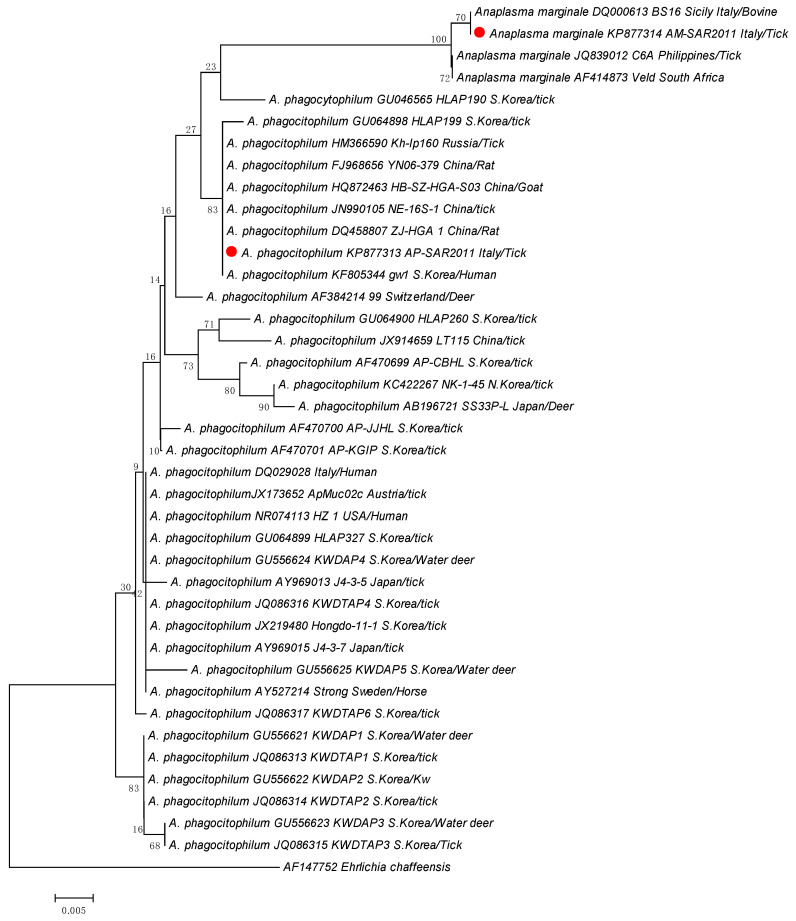
Phylogenetic tree based on 373-bp 16S rRNA gene of *A. phagocytophilum* and *A. marginale* strains collected in this study (red dots) compared with *Anaplasma* species obtained from GenBank database. The tree was constructed using the software MEGA6 and the neighbor-joining method with 1000 bootstrap re-samplings. *Ehrlichia chaffeensis* was used as outgroup. Country, hosts, and GenBank accession number are also indicated. Scale bar: number of base substitutions per site.

**Table 1 animals-13-01332-t001:** Species, sex, hosts, and geographic areas from which ticks were collected.

	N. of Ticks	Sex	Host (n.)	Collection Sites
**Ogliastra Province**				
*Rh. sanguineus* s.l.	97	40 males 57 females	Goat (18)	Talana
*Rh. bursa*	13	6 males 7 females	Goat (1)	Jerzu
	1	1 male	Goat (1)	Talana
**Sassari Province**				
*Rh. sanguineus* s.l.	23	19 males 4 females	Marten (1)	Bono
*Rh. bursa*	22	1 male 21 females	Cattle (1)	Villanova Monteleone

**Table 2 animals-13-01332-t002:** Detection and identification by PCR and sequencing of *A. phagocitophilum* and *A. marginale* in ticks collected in Sardinia, Italy.

Tick Species	N. of Positive Ticks and Relative Sex	Host	*Anaplasma* Identification	Strain	GenBank Accession Number
*Rh. sanguineus* s.l.	2♂—2♀	Goat	*A. phagocitophilum*	AP-SAR2011	KP877313
	1♂	Marten	*A. marginale*	AM-SAR2011	KP877314
*Rh. Bursa*	4♀	Goat	*A. phagocitophilum*	AP-SAR2011	KP877313
	1♀	Cattle	*A. marginale*	AM-SAR2011	KP877314

## Data Availability

Not applicable.

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
