# Peer review of "Anaplasma Species in Ticks Infesting Mammals of Sardinia, Italy"

_animals, 2023, doi:10.3390/ani13081332_

Round 1
Reviewer 1 Report
This is the first study to describe the presence of A. marginale in Sardinian ticks and of A. phagocytophilum in ticks of Rhiphicephalus genus, although their vector competence for these ticks is yet to be proven. At some places, English could have been better. I have the following comments. Line 98-99: Comment - Please provide a brief description of the basis of morphological identification of ticks mentioned in this study (Rh. sanguineus, Rh. bursa) in materials and method section. Line 112-114: Comment - As claimed that this is the first study to describe the presence of A. marginale in Sardinian ticks and of A. phagocytophilum in ticks of Rhiphicephalus genus. Please add the gel pictures (293 bps fragment of anaplasma species) which demonstrate the presence of Anaplasma in tick DNA samples along with positive and negative controls. Line 119-111: Instead of PBS, it should be 'lysis buffer' provided in the kit. Please provide the catalog number for the QIAgen DNA extraction kit. Line 205: Please remove 'it' Line 55-57: Comment - English should be improvedAuthor Response
This is the first study to describe the presence of A. marginale in Sardinian ticks and of A. phagocytophilum in ticks of Rhiphicephalus genus, although their vector competence for these ticks is yet to be proven.
At some places, English could have been better.
Response: Thank you for your comments. The document has been improved by adding more information, changing some sentences and also improving the English.
I have the following comments.
-Line 98-99: Comment - Please provide a brief description of the basis of morphological identification of ticks mentioned in this study (Rh. sanguineus, Rh. bursa) in materials and method section.
Response: Done
-Line 112-114: Comment - As claimed that this is the first study to describe the presence of A. marginale in Sardinian ticks and of A. phagocytophilum in ticks of Rhiphicephalus genus. Please add the gel pictures (293 bps fragment of anaplasma species) which demonstrate the presence of Anaplasma in tick DNA samples along with positive and negative controls.
Response: Added
-Line 119-111: Instead of PBS, it should be 'lysis buffer' provided in the kit. Please provide the catalog number for the QIAgen DNA extraction kit.
Response: Added
-Line 205: Please remove 'it'
Response: Removed
Line 55-57: Comment - English should be improved
Response: Rewritten as follows: “The genus Anaplasma has been especially studied for its pathogenicity in farm ani-mals since various species of Anaplasma such as A. marginale, A. ovis, and A. bovis along with A. phagocytophilum are regarded as one of the biggest threats to livestock [6]. In fact, these agents significantly affect animal productivity and cause considerable economic losses to farmers due to the reduction of reproductive performance, decreased milk and meat production, abortion and death of the infected animal [7]. However, the clinical iden-tification of infected animals is challenging because immunocompetent hosts do not ex-hibit symptoms while clinical signs ranging from subclinical infections with mild to high fever, anorexia, respiratory symptoms have been described in immunocompromised hosts [8]. Although anaplasmosis is rarely fatal, leukopenia and impaired neutrophil and lym-phocyte function of the bacteria can make animals more susceptible to life-threatening secondary infections [8].”
Reviewer 2 Report
The study describes the first isolation of A. phagocitophilum and A. marginale from Rh. sanguineus and Rh. bursa in Sardinia. The findings are meaningful and worthy of publication, but some parts of the article should be reviewed. Some sentences are not clear and there are a few citation errors.
The topics are at times repetitive and the conclusions is too generic. A future study of the seroprevalence of Anaplasma spp in livestock is not considered and dogs have not been taken into account for their potential role in the transmission of the disease to humans.
Moreover, the study did not identify the ecotypes of A. phagocitophilum, which is essential information to be able to define the zoonotic relevance. The knowledge of this data would be very important and would give more significance to the work.
I would suggest to carry out further research before publishing the article. My revisions are listed below
2 Delete small
13 Delete great
18-20. Rewrite the sentence
25- delete “domestic and wild mammal” and specify the number and the species examined ( 20 goats, 1 cattle and 1 marten)
30-31- rewrite the sentence: This is the first study that describes the presence of A. marginale and of A. ……..genus in Sardinia.
54 and 57: the bibliographic reference is not pertinent to what is stated. Please add a proper reference.
61: please add the missing reference
62-63: delete this sentence or rewrite the sentence in a different way, because the concept does not reflect the meaning of the cited article
65-66: A. ovis, A. capra and A. bovis aren’t mentioned in the cited bibliography; rewrite the sentence, in particular change “have spread” and “is yet to be discovered” and add the appropriate references.
71: the cited reference n. 8 is about A. phagocitophilum in humans, not in animals!
73: the cited reference n. 13 does not describe any of the subclinical manifestations that the authors report
81-85: The sentence isn’t clear, please rewrite
87: delate “wild and domestic” and leave only “from mammals”
98: Was the marten killed or found dead (compare with line 230)? In any case, it should be animal and not animals
195: It is unclear whether this is the first report in Sardinia (but this was already stated in the previous sentence). As reported in the study n. 25 “Molecular results confirmed the presence of A. marginale, T. annulata and T. equi in R. bursa ticks. A. marginale was found in ticks feeding in cattle, sheep, and goats, and among questing ticks.”, A. marginale in R. bursa ticks has already been demonstrated
217, 220 and 221: Please add the references
221-228: rewrite
229-231: remove this sentence - you cannot prove the cause of the marten’s death- or add a reference about the disease in this wild species.
237: Where was A. phagocitophilum detected for the first time? In Sardinia? Please add it
240-243: The sentence is not clear, please rewrite and add something about the role of migratory birds, as you mention them in line 84
262: Check the reference number (84)
279: It seems that the losses due to ruminant abortion is caused by A. phagocitophilum infection. Please rewrite the sentence and add some date to support the presence of A. phagocitophilum in Sardinian goats (i.e. serological screening)
323: reference n. 6 is not reported in the text
Author Response
The study describes the first isolation of A. phagocitophilum and A. marginale from Rh. sanguineus and Rh. bursa in Sardinia. The findings are meaningful and worthy of publication, but some parts of the article should be reviewed. Some sentences are not clear and there are a few citation errors.
Response: Thank you for your suggestion that help us to improve our manuscript. The paper has been reviewed following your suggestions and the citations have been corrected choosing those more appropriate.
The topics are at times repetitive and the conclusions is too generic. A future study of the seroprevalence of Anaplasma spp in livestock is not considered and dogs have not been taken into account for their potential role in the transmission of the disease to humans.
Response: These topics have been added in the conclusion
Moreover, the study did not identify the ecotypes of A. phagocitophilum, which is essential information to be able to define the zoonotic relevance. The knowledge of this data would be very important and would give more significance to the work. I would suggest to carry out further research before publishing the article. My revisions are listed below
Response: We thank the reviewer for the accuracy and care he put on revising our manuscript. We agree that the analysis of ecotypes would add precious information to the article. However, this article is a preliminary manuscript where the same methodology adopted for the detection of other Anaplasma species in ticks have been applied to this tick collection. We are now working on a more extended paper by using multiple genes to resolve phylogeny of these anaplasma strains detected in ticks, and as all results will be included in the next paper, we think that the information we provide here is sufficient for the screening done in this paper..
2 Delete small
Response: Deleted
13 Delete great
Response: Deleted
18-20. Rewrite the sentence
Response: the sentence has been rewritten as follows: “Ticks are hematophagous ectoparasites that are recognized for their ability to vector a wide variety of pathogens of viral, bacterial, protozoal, and helminthic nature to vertebrate hosts.”
25- delete “domestic and wild mammal” and specify the number and the species examined ( 20 goats, 1 cattle and 1 marten)
Response: Thanks for your comment, now the number and species of animals have been added.
30-31- rewrite the sentence: This is the first study that describes the presence of A. marginale and of A. ……..genus in Sardinia.
Response: Rewritten as follows: “In this study, we provide the first description and molecular detection of A. marginale and A. phagocytophilum in ticks of Rhiphicephalus genus in Sardinia.”
54 and 57: the bibliographic reference is not pertinent to what is stated. Please add a proper reference.
Response: The reference has been modified
61: please add the missing reference
Response: Added
62-63: delete this sentence or rewrite the sentence in a different way, because the concept does not reflect the meaning of the cited article
Response: Rewritten
“Among them, several species previously considered non-pathogenic, are recently considered zoonotic and associated with human diseases, suggesting that the number of Anaplasma species that are infecting humans is increasing.”
65-66: A. ovis, A. capra and A. bovis aren’t mentioned in the cited bibliography; rewrite the sentence, in particular change “have spread” and “is yet to be discovered” and add the appropriate references.
Response: Added and rewritten as follows:
“Among them, several species previously considered non-pathogenic, are recently considered zoonotic and associated with human diseases, suggesting that the number of Anaplasma species that are infecting humans is increasing. Specifically, A. phagocytophilum, the causative agent of tick-borne fever in sheep and granulocytic anaplasmosis in dogs (CGA), and horses (EGA) is also responsible for human infection (HGA). A. bovis, previously considered as infecting bovine monocytes, has been recently detected in humans in China [10]. Anaplasma platys, the etiological agent of infectious cyclic thrombocytopenia in dogs, that infects platelets has been documented in two women from Venezuela exposed to Rhipicephalus sanguineus [11]. A variant of the erythrocytic anaplasmal A. ovis was identified in a Cypriot patient with clinical signs including fever, hepatosplenomegaly and lymphadenopathy [12]. Finally, Anaplasma capra, a novel tick-borne pathogens, which was detected in China in 2010-2012, causes zoonotic infections and infects many different animal species, including humans [13].”
71: the cited reference n. 8 is about A. phagocitophilum in humans, not in animals!
Response: Rewritten
73: the cited reference n. 13 does not describe any of the subclinical manifestations that the authors report
Response: Rewritten
81-85: The sentence isn’t clear, please rewrite
Response: Rewritten: “Sardinia is the second biggest island in the Mediterranean Sea covering a surface of 24.090 km² and with different habitat types. It has an annual mean temperature of 22°C, and a typical Mediterranean climate that allows the survival of ticks during whole year. Furthermore, the island which is located approximately halfway between Spain, Italy and North Africa, is an important stopover area for migratory birds which pose a risk for the introduction and dispersal of ticks and TBD. Specific studies of prevalence of Anaplasma spp. in Sardinian ticks are few. It was previously observed that ticks belonging to Rhipicephalus and Hyalomma genera are the most frequent hosts for A. ovis, A. phagocytophilum, A. platys, and A. platys-like, it suggesting that these species could serve as potential vectors for these pathogens [14].”
87: delate “wild and domestic” and leave only “from mammals”
Response: Deleted
98: Was the marten killed or found dead (compare with line 230)? In any case, it should be animal and not animals
Response: Rewritten: “Ticks from one marten that was found dead, were provided by hunters who removed specimens from the animal.”
195: It is unclear whether this is the first report in Sardinia (but this was already stated in the previous sentence). As reported in the study n. 25 “Molecular results confirmed the presence of A. marginale, T. annulata and T. equi in R. bursa ticks. A. marginale was found in ticks feeding in cattle, sheep, and goats, and among questing ticks.”, A. marginale in R. bursa ticks has already been demonstrated
Response: Rewritten: “To the best of our knowledge, this is the first molecular evidence of A. marginale in Rh. bursa and Rh. sanguineus s.l. ticks collected from goats and a marten in Sardinia, a region in which species of the Rhipicephalus genus are widely distributed.
217, 220 and 221: Please add the references
Response: Added
221-228: rewrite
Response: It has been rewritten
“In the southern regions of Italy (Sicily, Puglia, Campania and Basilicata), where bovine anaplasmosis is endemic, the presence of A. marginale in Rh. turanicus and Haemaphysalis punctata collected from cattle has been previously reported [27].”
229-231: remove this sentence - you cannot prove the cause of the marten’s death- or add a reference about the disease in this wild species.
Response: The sentence now reads: “In this study, ticks that tested positive for A. marginale were removed from asymptomatic goats. However, we cannot know the health status of the marten whose ticks tested positive for A. marginale since it was found dead for unknown causes. Results from a previous study conducted in northeastern Hungary, highlighted the presence of Anaplasma sp. in spleen and liver samples of European pine marten in which the zoonotic ecotype I of A. phagocytophilum has been identified [31].
237: Where was A. phagocitophilum detected for the first time? In Sardinia? Please add it
Response: Added
“This study reports also the first molecular detection of A. phagocytophilum in Rh. sanguineus and Rh. bursa ticks in Sardinia indicating a potential role of these tick species in the epidemiology of the disease.”
240-243: The sentence is not clear, please rewrite and add something about the role of migratory birds, as you mention them in line 84
Response: The sentence has been reformulated as follows:
“It could be related to anthropogenic activities that contribute directly or indirectly in the emergence and re-emergence of tick-borne pathogens (e.g. animal production, animal–human interfacing, and globalization). Moreover, the role of migratory birds in the spread of ticks and their role in the circulation and dissemination of pathogens in Europe cannot be ruled out. During seasonal migrations, birds that cover short, medium or long distances within one or more distant geographical regions can carrier ticks and related pathogens introducing ticks and pathogenic species to areas where they have never occurred [41].”
262: Check the reference number (84)
Response: Deleted
279: It seems that the losses due to ruminant abortion is caused by A. phagocitophilum infection. Please rewrite the sentence and add some date to support the presence of A. phagocitophilum in Sardinian goats (i.e. serological screening)
Response: rewritten as follows: “Veterinarians should not overlook the presence of A. phogocytophilum in Sardinian goats and evaluation of the potential role of Anaplasma species as abortifacient agents should be also taken into account.”
323: reference n. 6 is not reported in the text
Response: Added
Round 2
Reviewer 1 Report
Thank you for making suggested changes.
Reviewer 2 Report
The manuscript has been sufficiently improved to warrant pubblication in Animals